# Game Design for Eliciting Distinguishable Behavior

**Fan Yang**[1*], **Liu Leqi**[1*], **Yifan Wu**[1*],
**Zachary C. Lipton**[1†], **Pradeep Ravikumar**[1*], **William W. Cohen**[1,2*], **Tom Mitchell**[1†]
[1]Carnegie Mellon University    [2] Google Inc.
*{fanyang1,leqil,yw4,pradeepr,wcohen}@cs.cmu.edu
†{zlipton, tom.mitchell}@cmu.edu

## Abstract

The ability to inferring latent psychological traits from human behavior is key to developing personalized human-interacting machine learning systems. Approaches to infer such traits range from surveys to manually-constructed experiments and games. However, these traditional games are limited because they are typically designed based on heuristics. In this paper, we formulate the task of designing *behavior diagnostic games* that elicit distinguishable behavior as a mutual information maximization problem, which can be solved by optimizing a variational lower bound. Our framework is instantiated by using prospect theory to model varying player traits, and Markov Decision Processes to parameterize the games. We validate our approach empirically, showing that our designed games can successfully distinguish among players with different traits, outperforming manually-designed ones by a large margin.

## 1  Introduction

Human behavior can vary widely across individuals. For instance, due to varying risk preferences, some people arrive extremely early at an airport, while others arrive the last minute. Being able to infer these latent psychological traits, such as risk preferences or discount factors for future rewards, is of broad multi-disciplinary interest, within psychology, behavioral economics, as well as machine learning. As machine learning finds broader societal usage, understanding users' latent preferences is crucial to personalizing these data-driven systems to individual users.

In order to infer such psychological traits, which require cognitive rather than physiological assessment (e.g. blood tests), we need an interactive environment to engage users and elicit their behavior. Approaches to do so have ranged from questionnaires [7, 17, 9, 24] to games [2, 10, 21, 20] that involve planning and decision making. It is this latter approach of game that we consider in this paper. However, there has been some recent criticism of such manually-designed games [3, 5, 8]. In particular, a game is said to be effective, or *behavior diagnostic*, if the differing latent traits of players can be accurately inferred based on their game play behavior. However, manually-designed games are typically specified using heuristics that may not always be reliable or efficient for distinguishing human traits given game play behavior.

As a motivating example, consider a game environment where the player can choose to `stay` or `moveRight` on a `Path`. Each state on the `Path` has a reward. The player accumulates the reward as they move to a state. Suppose players have different preferences (e.g. some might magnify positive reward and not care too much about negative reward, while others might be the opposite), but are otherwise rational, so that they choose optimal strategies given their respective preferences. If we want to use this game to tell apart such players, how should we assign reward to each state in the `Path`? Heuristically, one might suggest there should be positive reward to lure gain-seeking players and negative reward to discourage the loss-averse ones, as shown in Figure 1a. However, as shown

in Figure 1b and 2a, the induced behavior (either policy or sampled trajectories) are similar for players with different loss preferences, and consequently not helpful in distinguishing them based on their game play behavior. In Figure 1c, an alternative reward design is shown, which elicits more distinguishable behavior (see Figure 1d and 2b). This example illustrates that it is nontrivial to design effective games based on intuitive heuristics, and a more systematic approach is needed.

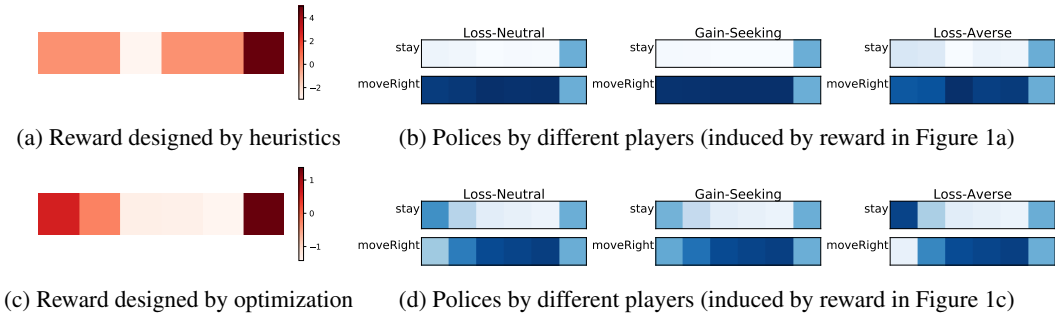

(a) Reward designed by heuristics   (b) Polices by different players (induced by reward in Figure 1a)

(c) Reward designed by optimization   (d) Polices by different players (induced by reward in Figure 1c)

Figure 1: Comparing reward designed by heuristics and by optimization. The game is a 6-state Markov Decision Process. Each state is represented as a square (see 1a or 1c) and player can choose `stay` or `moveRight`. The goal is to design reward for each state such that different types of players (Loss-Neutral, Gain-Seeking, Loss-Averse) have different behaviors. We show reward designed by heuristics in 1a and by optimization in 1c. Using these rewards, the policies of different players are shown on the right. For each player, its policy specifics the probability of taking an action (`stay` or `moveRight`) at each state. For example, Loss-Neutral's policy in 1d shows that it is more likely to choose `stay` than `moveRight` at the first (i.e. left-most) state, while in the second to fifth states, choosing `moveRight` has a higher probability.

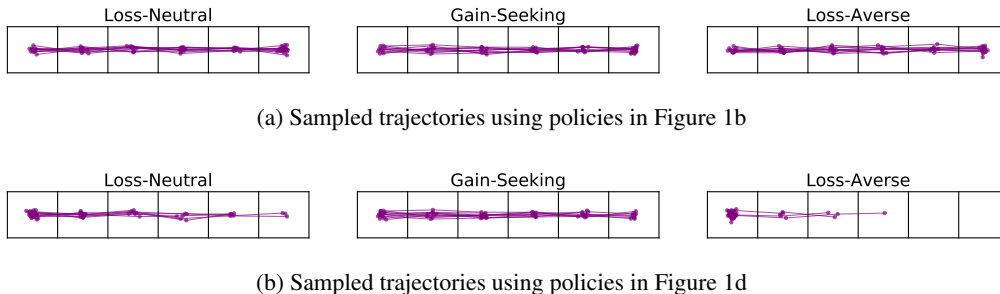

(a) Sampled trajectories using policies in Figure 1b

(b) Sampled trajectories using policies in Figure 1d

Figure 2: Comparing sampled trajectories using policies induced by different rewards. To further visualize how each type of players behave given different rewards, we sample trajectories using their induced policies. Given reward designed by heuristics (Figure 1a), all players behave similarly by traversing all the states (see 2a). However, given reward designed by optimization (Figure 1c), Gain-Seeking and Loss-Averse players behave differently. In particular, Loss-Averse chooses `stay` most of the time (see 2b), since the first state has a relatively large reward. Hence, reward designed by optimization is more effective at eliciting distinctive behaviors.

In this work, we formalize this task of *behavior diagnostic* game design, introducing a general framework consisting of a player space, game space, and interaction model. We use mutual information to quantitatively capture game effectiveness, and present a practical algorithm that optimizes a variational lower bound. We instantiate our framework by setting the player space using prospect theory [15], and setting the game space and interaction model using Markov Decision Processes [13]. Empirically, our quantitative optimization-based approach designs games that are more effective at inferring players' latent traits, outperforming manually-designed games by a large margin. In addition, we study how the assumptions in our instantiation affect the effectiveness of the designed games, showing that they are robust to modest misspecification of the assumptions.

## 2 Behavior Diagnostic Game Design

We consider the problem of designing interactive games that are informative in the sense that a player's type can be inferred from their play. A game-playing process contains three components: a player $z$, a game $\mathcal{M}$ and an interaction model $\Psi$. Here, we assume each player (which is represented by its latent trait) lies in some player space $\mathcal{Z}$. We also denote $Z \in \mathcal{Z}$ as the random variable corresponding to a randomly selected player from $\mathcal{Z}$ with respect to some prior (e.g. uniform) distribution $p_Z$ over $\mathcal{Z}$. Further, we assume there is a family of parameterized games $\{\mathcal{M}_\theta : \theta \in \Theta\}$. Given a player $z \sim p_Z$, a game $\mathcal{M}_\theta$, the interaction model $\Psi$ describes how a behavioral observation $x$ from some observation space $\mathcal{X}$ is generated. Specifically, each round of game play generates behavioral observations $x \in \mathcal{X}$ as $x \sim \Psi(z, \mathcal{M}_\theta)$, where the interaction model $\Psi(z, \mathcal{M}_\theta)$ is some distribution over the observation space $\mathcal{X}$. In this work, we assume $p_Z$ and $\Psi$ are fixed and known. Our goal is to design a game $\mathcal{M}_\theta$ such that the generated behavior observations $x$ are most informative for inferring the player $z \in \mathcal{Z}$.

### 2.1 Maximizing Mutual Information

Our problem formulation introduces a probabilistic model over the players $Z$ (as specified by the prior distribution $p_Z$) and the behavioral observations $X$ (as specified by $\Psi(Z, \mathcal{M}_\theta)$), so that $p_{Z,X}(z, x) = p_Z(z) \cdot \Psi(z, \mathcal{M}_\theta)(x)$. Our goal can then be interpreted as maximizing the information on $Z$ contained in $X$, which can be captured by the mutual information between $Z$ and $X$:

$$I(Z, X) = \int \int p_{Z,X}(z, x) \log \frac{p_{Z,X}(z, x)}{p_Z(z) p_X(x)} \, \mathrm{d}z \mathrm{d}x \tag{1}$$

$$= \int \int p_Z(z) \cdot \Psi(z, \mathcal{M}_\theta)(x) \log \frac{p_{Z|X}(z|x)}{p_Z(z)} \, \mathrm{d}z \mathrm{d}x, \tag{2}$$

so that the mutual information is a function of the game parameters $\theta \in \Theta$.

**Definition 2.1.** (Behavior Diagnostic Game Design) Given a player space $\mathcal{Z}$, a family of parameterized games $\mathcal{M}_\theta$, and an interaction model $\Psi(z, \mathcal{M}_\theta)$, our goal is to find:

$$\theta^* = \arg\max_\theta I(Z; X). \tag{3}$$

### 2.2 Variational Mutual Information Maximization

It is difficult to directly optimize the mutual information objective in Eq (2), as it requires access to a posterior $p_{Z|X}(z|x)$ that does not have a closed analytical form. Following the derivations in [6] and [1], we opt to maximize a variational lower bound of the mutual information objective. Letting $q_{Z|X}(z|x)$ denote any variational distribution that approximates $p_{Z|X}(z|x)$, and $\mathcal{H}(Z)$ denote the marginal entropy of $Z$, we can bound the mutual information as:

$$I(Z, X) = \int \int p_Z(z) \cdot \Psi(z, \mathcal{M}_\theta)(x) \log \frac{p_{Z|X}(z|x)}{p_Z(z)} \, \mathrm{d}z \mathrm{d}x \tag{4}$$

$$= \mathbb{E}_{z \sim p_Z} \left[ \mathbb{E}_{x \sim \Psi(z, \mathcal{M}_\theta)} \left[ \log \frac{p_{Z|X}(z|x)}{q_{Z|X}(z|x)} + \log q_{Z|X}(z|x) \right] \right] + \mathcal{H}(Z) \tag{5}$$

$$\geq \mathbb{E}_{z \sim p_Z, x \sim \Psi(z, \mathcal{M}_\theta)} \left[ \log q_{Z|X}(z|x) \right] + \mathcal{H}(Z), \tag{6}$$

so that the expression in Eq (6) forms a variational lower bound for the mutual information $I(Z, X)$.

## 3 Instantiation: Prospect Theory based MDP Design

Our framework in Section 2 provides a systematic view of *behavior diagnostic* game design, and each of its components can be chosen based on contexts and applications. We present one instantiation by setting the player space $\mathcal{Z}$, the game $\mathcal{M}$, and the interaction model $\Psi$. For the player space $\mathcal{Z}$, we use prospect theory [15] to describe how players perceive or distort values. We model the game $\mathcal{M}$ as a Markov Decision Process [13]. Finally, a (noisy) value iteration is used to model players' planning and decision-making, which is part of the interaction model $\Psi$. In the next subsection, we provide a brief background of these key ingredients.

## 3.1 Background

**Prospect Theory**   Prospect theory [15] describes the phenomenon that different people can perceive the same numerical values differently. For example, people who are averse to loss, e.g. it is better to not lose \$5 than to win \$5, magnify the negative reward or penalty. Following [23], we use the following distortion function $v$ to describe how people shrink or magnify numerical values,

$$v(r; \xi_{\text{pos}}, \xi_{\text{neg}}, \xi_{\text{ref}}) = \begin{cases} (r - \xi_{\text{ref}})^{\xi_{\text{pos}}} & r \geq \xi_{\text{ref}} \\ -(\xi_{\text{ref}} - r)^{\xi_{\text{neg}}} & r < \xi_{\text{ref}} \end{cases} \tag{7}$$

where $\xi_{\text{ref}}$ is the reference point that people compare numerical values against. $\xi_{\text{pos}}$ and $\xi_{\text{neg}}$ are the amount of distortion applied to the positive and negative amount of the reward with respect to the reference point.

We use this framework of prospect theory to specify our player space. Specifically, we represent a player $z$ by their personalized distortion parameters, so that $z = (\xi_{\text{pos}}, \xi_{\text{neg}}, \xi_{\text{ref}})$. In this work, unless we specify otherwise, we assume that $\xi_{\text{ref}}$ is set to zero. Given these distortion parameters, the players perceive a *distorted* $v(R; z)$ of any reward $R$ in the game, as we detail in the discussion of the interaction model $\Psi$ subsequently.

**Markov Decision Process**   A Markov Decision Process (MDP) $\mathcal{M}$ is defined by $(\mathcal{S}, \mathcal{A}, T, R, \gamma)$, where $\mathcal{S}$ is the state space and $\mathcal{A}$ is the action space. For each state-action pair $(s, a)$, $T(\cdot|s, a)$ is a probability distribution over the next state. $R : \mathcal{S} \to \mathbb{R}$ specifies a reward function. $\gamma \in (0, 1)$ is a discount factor. We assume that both states and actions are discrete and finite. For all $s \in \mathcal{S}$, a policy $\pi(\cdot|s)$ defines a distribution over actions to take at state $s$. A policy for an MDP $\mathcal{M}$ is denoted as $\pi_{\mathcal{M}}$.

**Value Iteration**   Given a Markov Decision Process $(\mathcal{S}, \mathcal{A}, T, R, \gamma)$, *value iteration* is an algorithm that can be written as a simple update operation, which combines one sweep of policy evaluation and one sweep of policy improvement [25],

$$V(s) \leftarrow \max_a \sum_{s'} T(s'|s, a)(R(s') + \gamma V(s')) \tag{8}$$

and computes a value function $V : \mathcal{S} \to \mathbb{R}$ for each state $s \in \mathcal{S}$. A probabilistic policy can be defined similar to maximum entropy policy [28, 18] based on the value function, i.e.,

$$\pi(a|s) = \operatorname*{softmax}_a \sum_{s'} T(s'|s, a)(R(s') + \gamma V(s')) \tag{9}$$

Value iteration converges to an optimal policy for discounted finite MDPs [25].

## 3.2 Instantiation

In this instantiation, we consider the game $\mathcal{M}_\theta := (\mathcal{S}, \mathcal{A}, T_\theta, R_\theta, \gamma)$, and design the reward function and the transition probabilities of the game by *learning* the parameters $\theta$. We assume that each player $z = (\xi_{\text{pos}}, \xi_{\text{neg}})$ behaves according to a noisy near-optimal policy $\pi$ defined by value iteration and an MDP with distorted reward $(\mathcal{S}, \mathcal{A}, T_\theta, v(R_\theta; z), \gamma)$. The game play has $L$ steps in total. The interaction model $\Psi(z, \mathcal{M}_\theta)$ is then the distribution of the trajectories $x = \{(s_t, a_t)\}_{t=1}^L$, where the player always starts at $s_{\text{init}}$, and at each state $s_t$, we sample an action $a_t$ using the Gumbel-max trick [11] with a noise parameter $\lambda$. Specifically, let the probability over actions $\pi(\cdot|s_t)$ be $(u_1, \ldots, u_{|\mathcal{A}|})$ and $g_i$ be independent samples from Gumbel$(0, 1)$, a sampled action $a_t$ can be defined as

$$a_t = \arg \max_{i=1,\ldots,|\mathcal{A}|} \frac{\log(u_i)}{\lambda} + g_i. \tag{10}$$

When $\lambda = 1$, there is no noise and $a_t$ distributes according to $\pi(\cdot|s_t)$. The amount of noise increases as $\lambda$ increases. Similarly, we sample the next state $s_{t+1}$ from $T(\cdot|s_t, a_t)$ using Gumbel-max, with $\lambda$ always set to one.

Our goal in *behavior diagnostic* game design then reduces to solving the optimization in Eq (3), where the player space $\mathcal{Z}$ consisting of distortion parameters $z = (\xi_{\text{pos}}, \xi_{\text{neg}})$, the game space parameterized

as $\mathcal{M}_\theta = (\mathcal{S}, \mathcal{A}, T_\theta, R_\theta, \gamma)$, and an interaction model $\Psi(z, \mathcal{M}_\theta)$ of trajectories $x$ generated by noisy value iteration using distorted reward $v(R_\theta; z)$,

As discussed in the previous section, for computational tractability, we optimize the variational lower bound from Eq (6) on this mutual information. As the variational distribution $q_{Z|X}$, we use a factored Gaussian, with means parameterized by a Recurrent Neural Network [12]. The input at each step is the concatenation of a one-hot (or softmax approximated) encoding of state $s_t$ and action $a_t$. We optimize this variational bound via stochastic gradient descent [16]. In order for the objective to be end-to-end differentiable, during trajectory sampling, we use the Gumbel-softmax trick [14, 19], which uses the softmax function as a continuous approximation to the argmax operator in Eq (10).

## 4 Experiments

### 4.1 Learning to Design Games

We learn to design games $\mathcal{M}$ by maximizing the mutual information objective in Eq (6), with known player prior $p_Z$ and interaction model $\Psi$. We study how the degrees of freedom in games $\mathcal{M}$ affect the mutual information. In particular, we consider environments that are `Path` or `Grid` of various sizes. `Path` environment has two actions, `stay` and `moveRight`. And `Grid` has one additional action `moveUp`. Besides learning the reward $R_\theta$, we also consider learning part of the transition $T_\theta$. To be more specific, we learn the probability $\alpha_s$ that the action `moveRight` actually stays in the same state. Therefore `moveRight` becomes a probabilistic action that at each state $s$,

$$p(s'|s, \texttt{moveRight}) = \begin{cases} \alpha_s & \text{if } s' = s \\ 1 - \alpha_s & \text{if } s' = s + 1 \\ 0 & \text{otherwise} \end{cases} \tag{11}$$

We experiment with `Path` of length 6 and `Grid` of size 3 by 6. The player prior $p_Z$ is uniform over $[0.5, 1.5] \times [0.5, 1.5]$. For the baseline, we manually design the reward for each state $s$ to be[1]

$$R_{\texttt{Path}}(s) = \begin{cases} -3 & \text{if } s = 3 \\ 5 & \text{if } s = 6 \\ 0 & \text{otherwise} \end{cases} \qquad R_{\texttt{Grid}}(s) = \begin{cases} -3 & \text{if } s = 9 \text{ (a middle state)} \\ 5 & \text{if } s = 18 \text{ (a corner state)} \\ 0 & \text{otherwise} \end{cases}$$

The intuition behind this design is that the positive reward at the end of the `Path` (or `Grid`) will encourage players to explore the environment, while the negative reward in the middle will discourage players that are averse to loss but not affect gain-seeking players, hence elicit distinctive behavior.

In Table 1, mutual information optimization losses for different settings are listed. The baselines have higher mutual information loss than other learnable settings. When only the reward is learnable, the `Grid` setting achieves slightly better mutual information than the `Path` one. However, when both reward and transition are learnable, the `Grid` setting significantly outperforms the others. This shows that our optimization-based approach can effectively search through the large game space and find the ones that are more informative.

Table 1: Mutual Information Optimization Loss

| Baselines | | Learn Reward Only | | Learn Reward and Transition | |
|---|---|---|---|---|---|
| Path (1 x 6) | Grid (3 x 6) | Path (1 x 6) | Grid (3 x 6) | Path (1 x 6) | Grid (3 x 6) |
| 0.111 | 0.115 | 0.108 | 0.099 | 0.107 | **0.078** |

## 4.2 Player Classification Using Designed Games

To further quantify the effectiveness of learned games, we create downstream classification tasks. In the classification setting, each data instance consists of a player label $y$ and its behavior (i.e. trajectory) $x$. We assume that the player attribute $z$ is sampled from a mixture of Gaussians.

$$z \sim \sum_{k=1}^{K} \frac{1}{K} \cdot \mathcal{N} \left( \begin{bmatrix} \xi_{\text{pos}}^k \\ \xi_{\text{neg}}^k \end{bmatrix}, 0.1 \cdot \mathcal{I} \right) \tag{12}$$

The label $y$ for each player corresponds to the component $k$, and the trajectory is sampled from $x \sim \Psi\left(z, \mathcal{M}_{\hat{\theta}}\right)$, where $\mathcal{M}_{\hat{\theta}}$ is a learned game. There are three types (i.e. components) of players, namely Loss-Neutral ($\xi_{\text{pos}} = 1, \xi_{\text{neg}} = 1$), Gain-Seeking ($\xi_{\text{pos}} = 1.2, \xi_{\text{neg}} = 0.7$), and Loss-Averse ($\xi_{\text{pos}} = 0.7, \xi_{\text{neg}} = 1.2$). We simulate 1000 data instances for train, and 100 each for validation and test. We use a model similar to the one used for $q_{Z|X}$, except for the last layer, which now outputs a categorical distribution. The optimization is run for 20 epochs and five rounds with different random seed. Validation set is used to select the test set performance. Mean test accuracy and its standard deviation are reported in Table 2.

Table 2: Classification Task Accuracy

| Baselines | | Learn Reward Only | | Learn Reward and Transition | |
|---|---|---|---|---|---|
| Path (1 x 6) | Grid (3 x 6) | Path (1 x 6) | Grid (3 x 6) | Path (1 x 6) | Grid (3 x 6) |
| 0.442 (0.056) | 0.482(.052) | 0.678 (0.044) | 0.658 (0.066) | 0.686 (0.044) | **0.822 (0.027)** |

Similar to the trend in mutual information, baseline methods have the lowest accuracies, which are about 35% less than any learned games. `Grid` with learned reward and transition outperforms other methods with a large margin of a 20% improvement in accuracy.

To get a more concrete understanding of learned games and the differences among them, we visualize two examples below. In Figure 3a, the learned reward for each state in a `Path` environment is shown via heatmap. The learned reward is similar to the manually designed one—highest at the end and negative in the middle—but with an important twist: there are also smaller positive rewards at the beginning of the `Path`. This twist successfully induces distinguishable behavior from Loss-Averse players. The induced policy (Figure 3b)) and sampled trajectories (Figure 3c)) are very different between Loss-Averse and Gain-Seeking. However, Loss-Neutral and Loss-Averse players still behave quite similarly in this game.

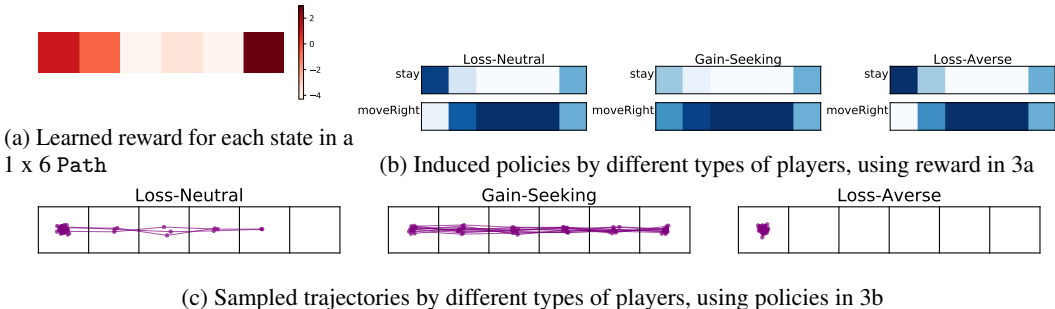

(a) Learned reward for each state in a 1 x 6 `Path`

(b) Induced policies by different types of players, using reward in 3a

(c) Sampled trajectories by different types of players, using policies in 3b

Figure 3: A 1 x 6 `Path` with learned reward. Gain-Seeking and Loss-Averse behave distinctively.

In Figure 4c and 4d, we show induced policies and sampled trajectories in a `Grid` environment where both reward and transition are learned. The learned game elicits distinctive behavior from different types of players. Loss-Averse players choose `moveUp` at the beginning and then always `stay`. Loss-Neutral players explore along the states in the bottom row, while Gain-Seeking players choose `moveUp` early on and explore the middle row. The learned reward and transition are visualized in Figure 4a and 4b. The middle row is particular interesting. The states have very mixed reward—some are relatively high and some are the lowest. We conjecture that the possibility of `stay` (when take `moveRight`) at some states with high reward (e.g. the first and third state from left in the middle row) makes Gain-Seeking behave differently from Loss-Neutral.

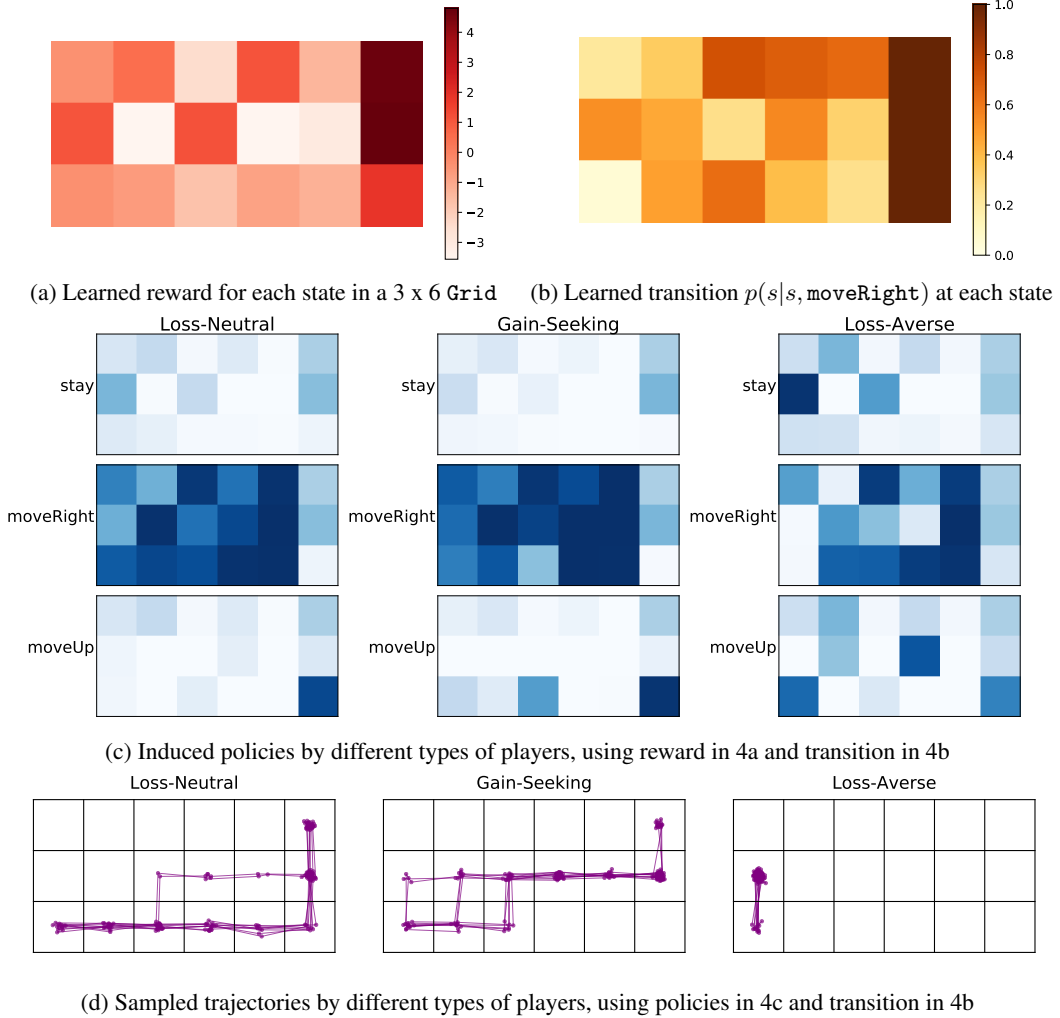

(a) Learned reward for each state in a 3 x 6 `Grid`     (b) Learned transition $p(s|s, \texttt{moveRight})$ at each state

(c) Induced policies by different types of players, using reward in 4a and transition in 4b

(d) Sampled trajectories by different types of players, using policies in 4c and transition in 4b

Figure 4: A 3 x 6 `Grid` with learned reward and transition (see 4a and 4b) elicit distinguishable behaviors from different types of players.

We also consider the case where the interaction model has noise, as described in Eq (10), when generating trajectories for classification data. In practice, it is unlikely that one interaction model describes all players, since players have a variety of individual differences. Hence it is important to study how effective the learned games are when downstream task interaction model $\Psi$ is noisy and deviates from assumption. In Table 3, classification accuracy on test set is shown at different noise level. We consider three designs here. As defined above, a baseline method which uses manually designed reward in `Path`, a `Path` environment with learned reward, and a `Grid` environment with both learned reward and transition. Interestingly, while adding noise decreases classification performance of learned games, the manually designed game (i.e. baseline method) achieves higher accuracy in the presence of noise. Nevertheless, the learned `Grid` outperforms others, though the margin decreases from 20% to 12%.

Table 3: Classification Accuracy at Different Noise Level

|  | $\lambda = 1$ | $\lambda = 1.5$ | $\lambda = 2.5$ |
|---|---|---|---|
| Path (1 x 6, Baseline) | 0.442 (0.056) | 0.510 (0.053) | 0.482 (0.041) |
| Path (1 x 6, Learn Reward Only) | 0.678 (0.044) | 0.678 (0.039) | 0.650 (0.048) |
| Grid (3 x 6, Learn Reward and Transition) | 0.822 (0.027) | 0.778 (0.044) | 0.730 (0.061) |

In Figure 5, we visualize the trajectories when the noise in the interaction model $\Psi$ is $\lambda = 2.5$. This provides intuition for why the classification performance decreases, as the boundary between the behavior of Loss-Neutral and Gain-Seeking is blurred.

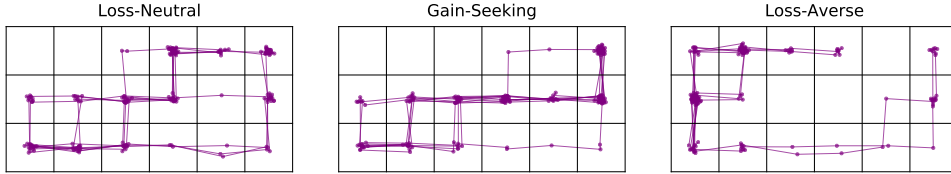

Figure 5: Sampled trajectories with noise $\lambda = 2.5$

### 4.3 Ablation Study

Lastly, we consider using different distributions for player prior $p_Z$ on $(\xi_{\text{pos}}, \xi_{\text{neg}})$, which could be agnostic of the downstream tasks or not. We compare the classification performance when $p_Z$ is uniform or biased towards the distribution of player types. We consider two cases: Full and Diagonal. In Full, the player prior $p_Z$ is uniform over $[0.5, 1.5] \times [0.5, 1.5]$. In Diagonal, $p_Z$ is uniform over the union of $[0.5, 1] \times [1, 1.5]$ and $[1, 1.5] \times [0.5, 1]$, which is a strict subset of the Full case and arguably closer to the player types distribution in the classification task. In Table 4, we show performance of games learned with Full or Diagonal player prior.

Table 4: Comparison of Learned Games with Different Player Prior $p_Z$

| Method | | Mutual Information Loss | | Classification Accuracy | |
|---|---|---|---|---|---|
| | | Full | Diagonal | Full | Diagonal |
| Reward Only | Path | 0.108 | 0.043 | 0.678 (0.044) | 0.658 (0.034) |
| Reward Only | Grid | 0.099 | 0.039 | 0.658 (0.066) | 0.662 (0.060) |
| Reward and Transition | Path | 0.107 | 0.043 | 0.686 (0.044) | 0.668 (0.048) |
| Reward and Transition | Grid | 0.078 | 0.036 | 0.822 (0.027) | 0.712 (0.051) |

Across all methods, using the Diagonal prior achieves lower mutual information loss compared to using the Full one. However, this trend does not generalize to classification. Using the Diagonal prior hurts classification accuracy. We visualize the sampled trajectories in Figure 6. As we can see, Loss-Neutral no longer has its own distinctive behavior, which is the case using Full prior (see Figure 4d). It seems that learned game is more likely to overfit the Diagonal prior, which leads to poor generalization on downstream tasks. Therefore, using a play prior $p_Z$ agnostic to downstream task might be preferred.

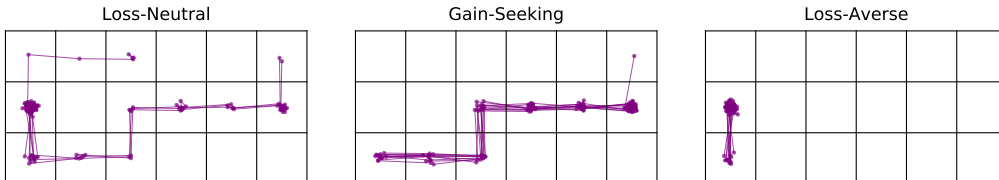

Figure 6: Sampled trajectories using Diagonal player prior

## 5 Conclusion and Discussion

We consider designing games for the purpose of distinguishing among different types of players. We propose a general framework and use mutual information to quantify the effectiveness. Comparing with games designed by heuristics, our optimization-based designs elicit more distinctive behaviors.

Our behavior-diagnostic game design framework can be applied to various applications, with player space, game space and interaction model instantiated by domain-specific ones. For example, [22] studies how to generate games for the purpose of differentiating players using player performance instead of behavior trajectory as the observation space. In addition, we have considered the case when the player traits inferred from their game playing behavior are stationary. However, as pointed out by [26, 27, 4], there can be complex relationships between players' in-game and outside-game personality traits. In future work, we look forward to addressing this distribution shift.

**Acknowledgement**   W.C. acknowledges the support of Google. L.L. and P.R. acknowledge the support of ONR via N000141812861. Z.L. acknowledges the support of the AI Ethics and Governance Fund. This research was supported in part by a grant from J. P. Morgan.

## Footnotes

[1]We have also experimented with different manually designed baseline reward functions. Their performances are similar to the presented baselines, both in terms of mutual information loss and player classifier accuracy. The performance of randomly generated rewards is worse than the manually designed baselines.

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
