[Reviews · NeurIPS 2019]

Reviewer 1



This is an interesting paper, which focuses on finding more effective ways to design better game settings that would be able to probe behavioural differences more effectively. This idea certainly has great potential, although it's possible that once we consider more complicated games, like the ones people play in reality (as opposed to toy games, which are regrettably still widely used in many behavioural and neuroeconomics studies), they already have sufficient dimensionality and flexibility to dissect different behavioural phenotypes. However, as this transition to digital phenotyping/ecological validity won't happen in a year (or even a decade), the presented approach still has potential to remain useful for years to come. The paper is generally well written and of good quality, although it could be clearer at places. Even though I understand that this is a methodological study, the fact that there is no behavioural validation to test if people indeed perform as predicted by the model and their behavioural parameters can be more effectively identified in the discovered game settings is a serious limitation to study's significance. I also have a number of more specific comments: - it's not clear why this particular "baseline" has been chosen. It doesn't seem like a particularly good/logical design for me - was it used in some actual behavioural economics studies? - the behavioural simulation/evaluation process is not entirely clear - is that a repetitive game design, i.e. do participants learn what they should and should not do from previous instances of the game and can improve with time? Otherwise, it's hard to understand how e.g. in Fig. 4 the loss averse group never goes right - how do they know these values are poor if they never visited them? - In prospect theory, loss aversion (which seems the main focus here) is normally modelled with a multiplier, whereas risk aversion/utility curvature is modelled as a power function - here it seems you try to do both using the latter, which slightly simplifies the model, but it would be interesting to see if the results would still be qualitatively the same if my mentioned formalisation were used (e.g. if utility curvatures are the same, only positive and negative slopes differ). My feeling is that some of the results may be due to utility curvature not due to loss-aversion/gain-preference per se. - Policy function should be explained better. For example, how does Gumbel-max compare to softmax (which I think is more common in behavioural and neuroeconomics). - Finally discussion should ideally be expanded, perhaps at a cost of moving some math/derivations to the supplementary material (although not leading to fewer explanations).

Reviewer 2



- The paper proposes a method to automatically design interactive games so that player with different latent preferences (e.g., reward seeking versus loss aversion) are easily separable from observing their action choices. Designing the reward functions for these games manually or heuristically is non-trivial as even simple behaviors (reward seeking versus loss aversion) are not easily separated by certain seemingly correct choices of reward functions as shown in Figure 1. The paper formalizes this notion of "behavior diagnostic game design". - The optimization problem of finding the game parameters is set up as the maximization of the mutual information between the player (z) and the observation (x) produced by z interacting with a chosen game. The exact problem is intractable so a variational bound is derived. - The player space p(z) is defined using a prospect theory model which models the distorted reward perceived by a player given their latent preferences as a triple of three numbers. Now assuming the player behaves near-optimally wrt to their reward, an MDP can be induced where the reward and transition functions are to be selected such that the variational lower bound of the mutual information objective is optimized. The implementation involves training a simple RNN to learn the variational distribution as the means of the factored gaussians. Overall, the formulation seems quite elegant to me. - The experiments involve path and grid domains and primarily involve the study of three behaviors: loss-neutral, loss-averse and gain-seeking. The reward and transition functions are learned in these simple environments. The learned reward and transition function correspond to the optimal parameters for maximizing the variational lower bound of the mutual information under the parameterization of the observation space and player space. The approach allows an effective search through the induced game space in order to identify (parameterized) games that produce higher mutual information. Larger game spaces produce better MI suggesting the search is working. Here, I'd be curious to see how different choices of baseline score. Specifically, how much variance is there in the MI as a function of the choice of R(s) in Eq 12? Some experiments with additional values of R(s) would be interesting to see. As it stands, the gap between methods seems a bit small numerically (0.107 vs 0.111) and I'm unable to tell if the difference is significant. Also, I don't understand why there is no baseline for the Grid environment. - The next set of experiments involve parameterizing the player space as a mixture of K (=3) Gaussians, where the three components correspond to different types of player behavior. The player labels (component) induce a classification task and here again the Grid environment shows the best ability to separate the player behaviors when a version of the model augmented for classification is used to predict player labels. Here the win over the baseline is much clearer. As before, a larger collection of player baselines may help illuminate the utility of the search a bit better. - The experiments also include studying the effect of noise as well as the effect of the priors. Overall, while the experiments do a good job of analyzing the proposed method, I think a more rigourous set of experiments with baselines would strengthen this portion of the paper. - Also, are human latent preferences limited to the 3 types considered here? Perhaps a discussion of why these 3 were considered and other possible behaviors (e.g., risk seeking?) might make the paper more accessible to a larger audience. - Overall, the paper contains an elegant formulation and solution to the problem of designing games to elicit separable human behaviors. Although I'm not very familiar with this body of work, I was able to understand the paper and its findings. The paper is exceptionally well written and the ideas flow very naturally. Overall, this seems like a very nice application of modern ML techniques to an application in cognition / psychology. I'm happy to recommend acceptance. UPDATE: The authors have satisfactorily addressed the points raised by the reviewers. My score remains unchanged.

Reviewer 3



I like the direction this research is going, and as far as I can see the approach is novel and sound and the results are good. It should be noted that the "games" designed here are very simple, game-theoretic games more than games that people actually play. Interestingly, there has been considerable work on using optimization methods to design games or game content, and also on characterizing player styles, but more more complex games. What you call "player preferences" are otherwise referred to as "procedural personas" (also motivated by prospect theory), e.g. in the work of Christoffer Holmgård: Holmgard, Christoffer, et al. "Automated playtesting with procedural personas with evolved heuristics." IEEE Transactions on Games (2018). These procedural personas have also been used for design optimization, so automatically generating levels that fit particular personas: Liapis, Antonios, et al. "Procedural personas as critics for dungeon generation." European Conference on the Applications of Evolutionary Computation. Springer, Cham, 2015. Games have also been optimized to be maximally discriminative between different players, for example: Nielsen, Thorbjørn S., et al. "Towards generating arcade game rules with VGDL." 2015 IEEE Conference on Computational Intelligence and Games (CIG). IEEE, 2015. It's important to note that in-game preferences and personas are not the same as preferences and personas outside of the game (the introduction to your paper seems to make that assumption). However, it is possible to model aspects of out-of-game stable traits from in-game behavior - the relation is complex. Some examples: Tekofsky, Shoshannah, et al. "Psyops: Personality assessment through gaming behavior." In Proceedings of the International Conference on the Foundations of Digital Games. 2013. Yee, Nick, et al. "Introverted elves & conscientious gnomes: the expression of personality in world of warcraft." Proceedings of the SIGCHI Conference on Human Factors in Computing Systems. ACM, 2011. Canossa, Alessandro, Josep B. Martinez, and Julian Togelius. "Give me a reason to dig Minecraft and psychology of motivation." 2013 IEEE Conference on Computational Inteligence in Games (CIG). IEEE, 2013. Hopefully this body of work could help you give some context for your own work.

[Author Response · NeurIPS 2019]

We thank all three reviewers for their detailed and thoughtful reviews. We were glad to see a consensus that the paper was "well-written" (R1, R2) and constituted an interesting interdisciplinary research direction (R1, R2, R3). Below, we address each reviewer's specific questions in turn.

**Reply to R1**

***"Why this baseline? Use baseline from literature?"*** We created this baseline guided by the intuition that by setting a high positive reward at the end of the path, with a negative reward along the route, we might identify those users willing vs unwilling to endure the obstacle. We are currently exploring ways to adapt pre-existing approaches from the prior literature to our problem. However, we note that because much of the related work does not address MDPs, adapting these methods faithfully to our setting can be non-trivial.

***"How about if the slopes differ?"*** Per your feedback, we ran new experiments where the slopes differ. In this setup, loss-neutral players have both slopes as 1, gain-seeking players slopes 2 (positive) and .5 (negative), and loss-averse slopes are .5 and 2, respectively. In the Path setting, these experiments yielded qualitatively similar results. We will add this analysis to the final version. With these players, the baseline achieves mutual information loss of .922, classification accuracy of .474 (.051 std dev). While the design optimized by our proposed framework has lower mutual information loss, .735, and much higher classification accuracy of .772 (.037 std dev).

***"Do the players learn from previous experience?"*** We do not model the player's learning but plan to in future work.

***Re: Gumbel-max and softmax*** Indeed our player policies *do* use softmax to produce a distribution over actions (Eq 9). Gumbel-max is a way to sample from this distribution with injected noise. Gumbel-softmax provides a continuous approximation to this *sampling step*, enabling end-to-end learning. We will clarify this in the revised text.

**Reply to R2**

***"A larger evaluation of the baselines?"*** To understand the variance of mutual information loss and classification accuracy for different choices of baseline, we considered three more baseline rewards where $r_{pos}$ and $r_{neg}$ vary. As a brief reminder, the baseline is defined as $R(s) = r_{neg}$ if $s = 3$ and $r_{pos}$ if $s = 6$, otherwise zero. For $(r_{pos}, r_{neg}) = (-3, 3)$, $(-5, 5)$, or $(-5, 3)$, the mutual information loss is .133, .111, .133, respectively. The classification accuracy (and std dev) is .472(.054), .460(.058), .492(.047), respectively. We also experimented with five more random baselines, where each state in one such baseline is a random number between $-5$ and 5. The averaged mutual information loss is $.150 \pm 0.02$ and classification accuracy is $.427 \pm 0.11$. Different baselines achieve similar performance with small variation. This indicates that search in the game space is necessary.

***"Why no baseline for the Grid? setting"*** Per your suggestion, we added a baseline for a Grid of size $3 \times 6$ where the last state ($s = 18$) has a positive reward 5 and the middle state ($s = 9$) has a negative reward $-3$, all other states have zero reward. Comparing with the reward found by optimization, this baseline has higher mutual information loss .115 and lower classification accuracy .482(.052).

***"Why these 3 types of human?"*** We chose these 3 types based on the risk preference patterns discussed in the seminal paper on prospect theory by Kahneman and Tversky. Among the many applications of these patterns, loss aversion has been widely studied in behavioral economics, e.g., Benartzi and Thaler [1995].

**Reply to R3**

***Re: contextualizing our work with respect to related literature*** Thanks for this rich list of related papers and for highlighting important connections that escaped our attention. We will incorporate these references, with discussion, in the final version. Notably, Nielsen et al. also study how to generate games for the purpose of differentiating players, but use a different objective function based on *Relative Algorithm Performance Profiles*. In contrast, our framework optimizes mutual information to distinguish player types. Reading Tekofsky et al., Yee et al., Canossa et al., we learned that there can be complex relationships between players' in-game and outside-game personality traits, which we assume are the same in this current work. In future work, we look forward to addressing this distribution shift.

# References

Shlomo Benartzi and Richard H Thaler. Myopic loss aversion and the equity premium puzzle. *The Quarterly Journal of Economics*, 1995.

Alessandro Canossa, Josep B Martinez, and Julian Togelius. Give me a reason to dig minecraft and psychology of motivation. In *Conference on Computational Intelligence in Games (CIG)*, 2013.

Thorbjørn S Nielsen, Gabriella AB Barros, Julian Togelius, and Mark J Nelson. Towards generating arcade game rules with vgdl. In *2015 IEEE Conference on Computational Intelligence and Games (CIG)*, pages 185–192. IEEE, 2015.

Shoshannah Tekofsky, Jaap Van Den Herik, Pieter Spronck, and Aske Plaat. Psyops: Personality assessment through gaming behavior. In *International Conference on the Foundations of Digital Games*, 2013.

Nick Yee, Nicolas Ducheneaut, Les Nelson, and Peter Likarish. Introverted elves & conscientious gnomes: the expression of personality in world of warcraft. In *Conference on Human Factors in Computing Systems (CHI)*, 2011.


[Meta-Review · NeurIPS 2019]

The three reviewers agreed on the quality and interest of the approach and results presented. Although there are no strong contributions of broad interest in this submission, I find this is a nice and well executed application paper with certain impact in game design. I recommend it is accepted as a poster. [This meta-review was reviewed and revised by the Program Chairs]